# Development and validation of a ground-based Spatial Heterodyne Spectroscopy **Asymmetric** (ASHS) system for sounding neutral wind in the mesopause

- Guangyi Zhu<sup>1,3</sup>, Yajun Zhu<sup>1,2,3,\*</sup>, Martin Kaufmann<sup>4</sup>, Tiancai Wang<sup>1,2,3</sup>, Weijun Liu<sup>1,3</sup>, Wei Yuan<sup>1,3</sup>, Siyin Liu<sup>1,2</sup>, Guotao Yang<sup>1,2,3</sup>, and Jiyao Xu<sup>1,2,3,\*</sup>
  - <sup>1</sup> State Key Laboratory of Solar Activity and Space Weather, National Space Science Center, Chinese Academy of Sciences, Beijing 100190, China
  - <sup>2</sup> University of Chinese Academy of Sciences, Beijing 100190, China
- 10 <sup>3</sup> Hainan National Field Science Observation and Research Observatory for Space Weather, Hainan 571734, China
  - <sup>4</sup> Institute of Energy and Climate Research (IEK-7), Jülich Research Centre, Jülich 52425, Germany

Correspondence to:

\* y.zhu@swl.ac.cn



\* <u>jyxu@spaceweather.ac.cn</u>

**Abstract.** Winds in the mesopause region are key to understanding atmospheric dynamics. This study presents a novel ground-based Asymmetric Spatial Heterodyne Spectroscopy (ASHS) system designed to measure these winds by observing the nighttime green line airglow of atomic oxygen. The system's configuration, thermal behavior, and calibration procedures are detailed. The as-built configuration meets the performance expectations established during the design phase, enabling wind measurements with an accuracy better than 2 m/s. Field observations from Mohe Station (53.5°N, 122.3°E) during geomagnetically quiet conditions demonstrate good agreement with co-located LiDAR data, validating the ASHS system's capability to derive 25 mesopause winds from interferograms.

### Plain language summary.

Winds in the mesopause region (85-100 km altitude) drive upper-atmospheric dynamics and energy transfer. We present the Asymmetric Spatial Heterodyne Spectrometer, a ground-based instrument, to measure winds by observing the green airglow of atomic oxygen. Lab tests demonstrated the instrument achieves better than 2 m/s accuracy. Field measurements at a highlatitude site in China showed strong agreement with independent LiDAR data, confirming that the system delivers reliable wind retrievals.

#### 1. Introduction





The Earth's upper mesosphere and lower thermosphere serve as a critical interface where chemical and physical processes are influenced by energy transmission from both the lower atmosphere and near-Earth space (Hedin et al., 1989; Kelly, 2009). Neutral winds are essential for understanding the dynamics, transport, and energy budget in the mesopause region, making them a key parameter in atmospheric models (Christensen et al., 2009; Dhadly et al., 2023).

Over the past few decades, optical passive sounding techniques have become pivotal for precise measurements of neutral winds in the middle and upper atmosphere. Utilizing these techniques, Doppler winds are accurately determined by analyzing airglow emission lines from particles transported by atmospheric winds. Among these instruments, the Fabry-Perot Interferometer (FPI) is particularly noteworthy, providing high-precision Doppler shift measurements using a Fabry-Perot etalon. Equally important is the Michelson Interferometer (MI), which can be field-widened to obtain a high etendue, delivering outstanding signal-to-noise ratios and detection efficiency. After decades of refinement, both FPI and MI have achieved widespread adoption in ground-based and satellite-based wind measurement applications (Hays et al., 1981; Gualt et al., 1996; Killen et al., 2006; Mclandress et al., 1996). Despite the valuable observations provided by these instrument, neutral wind measurements remain sparse over long-term periods and across wide areas.

A notable variant of the Michelson Interferometer is one that utilizes a technology called Asymmetric Spatial Heterodyne Spectroscopy (ASHS), first proposed in 2007 (Englert et al., 2007). The ASHS is an inherent concept of spatial heterodyne spectroscopy (SHS) by extending the optical path of one interferometer arm to observe interferogram at high optical path

differences (OPDs), which allows field-widening technology and interferometric measurements without moving parts while maintaining spectral resolution. Field-widening refers to modifying a conventional interferometer to enhance its ability to collect light over a wider field of view while maintaining its modulation efficiency. One application of this technology is the Redline DASH Demonstration Instrument (REDDI), which was built to make ground-based neutral wind observations by sounding atomic oxygen red line emission at 630 nm in thermosphere (Englert et al., 2010). Subsequently, a DASH interferometer, the Michelson Interferometer for Global High-Resolution Thermospheric Imaging (MIGHTI), was developed for the Ionospheric Connection Explorer (ICON) satellite to measure neutral winds by observing atomic oxygen green line emission at 557.7 nm and redline emission at 630.0 nm simultaneously in mesopause and thermosphere at an altitude of 90-300 km (Englert et al., 2017).

The cost-effective and precise ASHS instruments are ideally suited for establishing a ground-based network for thermospheric neutral wind observations. Compared with traditional MI, ASHS avoids complex scanning structures. Moreover, in comparison with MI and FPI, it can synchronously detect calibration spectral lines to mitigate the impact of instrumental drift. Wei et al. (2020) proposed a ground-based, thermally-insensitive monolithic ASHS interferometer for observing the atomic oxygen airglow at 630.0 nm. This configuration was verified in the laboratory and demonstrated excellent thermal stability (Dötzer, 2019; Liu et al., 2018; Wei et al., 2022). The methodology and design tools developed for that instrument are also adopted in the present work.

To the authors' knowledge, existing ground-based ASHS instruments are limited to observing the atomic oxygen red line in the middle and upper thermosphere. To extend wind measurements to a lower altitude region, it is advantageous to additionally observe the green line, which has its intensity peak just above the mesopause at around 96 km and spanning a layer approximately 10 km thick (Zhang and Shepherd, 2005). Detecting this faint emission

from the ground is challenging due to the low layer's thinness. The scattering of artificial light sources in the lower atmosphere can reduce the sensitivity and precision of the detection.

Several ground-based instruments have been developed to measure neutral winds in mesopause region by observing the atomic oxygen green line. For instance, a three-channel FPI has been developed by Wu et al. (2004) to detect the green line in mesopause region. Another instrument is a variant of the Michelson Interferometer, named the E-Region Wind Interferometer (ERWIN II), which facilitates the detection of the green line in mesopause region (Kristoffersen et al., 2013). For a ground-based ASHS system, where constraints such as mass and volume are less critical, combining multiple channels within a single interferometer is less essential. Instead, separate instruments can be individually optimized and tailored to the geometry and radiative properties of their respective emission layers to improve observation quality and temporal resolution.

This work focuses specifically on the design, calibration, and verification of an ASHS instrument dedicated to observing the atomic oxygen green line and is organized as follows: Firstly, the concept of the ASHS interferometer and the principle of wind measurements are described in Section 2. In Section 3, the optical design principle is discussed. In the subsequent section, the ASHS instrument configuration and main parameters are presented. In order to evaluate the performance of the instrument, a thermal stability test and wind measurement calibration are carried out in Section 5. In Section 6, horizontal wind is retrieved from field observations and compared with co-located LiDAR measurements. Section 7 provides a summary of the study.

### 2. ASHS concept






Fig. 1 illustrates the configuration of a field-widening ASHS interferometer. The incident light is split into two beams by a beamsplitter, which then reflect off two gratings and recombine to form interference fringes, which can be modeled according to the characterization of wavefronts (Liu et al., 2018). The fringe intensity I can be expressed as

$$I(x) \propto 1 + V_I(x) \cos[2\pi L(\sigma_0 - \sigma_L)] \exp[-2\pi^2 \sigma_D^2 L^2],$$
 (1)

where  $V_I(x)$  is the visibility function of the interferometer, which is a function of sampling position x on the detector,  $L = 4 \tan \theta \, x + 2 \Delta d$  is the OPD,  $\theta$  is the Littrow angle corresponding to Littrow wavenumber  $\sigma_L$ ,  $\sigma_0$  is the central wavenumber of the emission line, the term  $2\Delta d$  represents the optical path difference between the nominal and extended arms of the interferometer at the Littrow wavelength, and  $\sigma_D$  is the Doppler broadening parameter, which is defined as




$$\sigma_D = \sigma_0 \sqrt{\frac{k_B T}{mc^2}},\tag{2}$$

where,  $k_B$  is the Boltzmann constant, T is the kinetic temperature, m is the mass of the emitter and c is the velocity of light. The cosine term is the phase of fringe in the interferogram. The exponential term damps the visibility of the interferogram due to Doppler broadening of the emission lines. Accordingly, the phase term of fringes in the interferogram can be described as a function of the sampling position x

$$\phi(x) = 2\pi (4 \tan \theta x + 2\Delta d)(\sigma_0 - \sigma_L). \tag{3}$$

The phase difference between reference and Doppler-shifted interferograms can be expressed as:

$$\Delta\phi(x) = \phi'(x) - \phi(x) = 2\pi(4\tan\theta \, x + 2\Delta d)(\sigma_v - \sigma_0),\tag{4}$$

where  $\sigma_v$  is the Doppler-shifted wavenumber of the emission line. Then the Doppler velocity v can be determined from the phase difference  $\Delta \phi$  of two observations by

$$v = \frac{\Delta\phi \cdot c}{2\pi\sigma_0(4\tan\theta \, x + 2\Delta d)}.\tag{5}$$

Fig. 1. The configuration of the field-widening ASHS interferometer.

## 3. Optical design

### 3.1 Optical layout of ASHS interferometer

The configuration of the ASHS interferometer is similiter to the design of Wei et al. (2020), as shown in Fig. 2. Here  $t_0$  represents the nominal thickness of the beamsplitter, while  $t_1$  refers to the additional asymmetric extension in thickness,  $t_2$  is the thickness of the prism,  $d_1$  and  $d_2$  are the thickness of spacer 1 and spacer 2, respectively.

Fig. 2. The configuration of a thermally compensated field-widening ASHS interferometer. The extra optical path is introduced by extending one arm of the beamsplitter. The specifications of the ASHS Interferometer are listed in Table 1.

### 3.2. OPD offset



The choice of OPD offset is a balance of the visibility of the fringe pattern and the measurement sensitivity of the instrument. According to Eq. (1) the visibility can be expressed by an envelope function

$$Visbility = V_I(x) \exp(-2\pi^2 \sigma_D^2 L^2). \tag{6}$$

The visibility decreases with the OPD offset L. Conversely, the sensitivity of the ASHS interferometer to wind-induced phase shifts decreases as the OPD offset becomes smaller (see Eq. (5)).

A series of numerical simulations were conducted to determine the optimal OPD offset for measuring winds using the atomic oxygen green line emission at 557.7 nm. The interferogram of the atomic oxygen green line is simulated according to Eq. (1), including background radiation, random noise and transmission efficiency. The Doppler velocity is assumed to be 100 m/s and temperature is 200 K, which are typical values in the mesopause region (Englert et al.,

2023; García-Comas et al., 2008; Hall et al., 2006; Holmen et al., 2016; Yi et al., 2021). Fig.3 (a) shows the visibility and sensitivity variation when OPD offset is changed from 0 mm to 200 mm. The visibility remains relatively constant up to an OPD offset of 40 mm, then decreases approximately linearly up to 120 mm, after which it stabilizes again. In contrast, the phase difference increases linearly across the entire simulation range. By retrieving Doppler velocity from the phase difference, the wind measurement error corresponding to OPD offset can be estimated as shown in Fig. 3 (b). There are four lines representing wind retrieval errors under four different visibility conditions, with each line being the result of a Monte Carlo simulation with 10,000 samples. The wind uncertainty decreases rapidly from an OPD offset of 0 to 50 mm, forms a wide trough between 50 and 90 mm, and then increases again for larger OPD offsets. To minimize instrument size while reducing wind estimation error, an OPD offset of 55 mm in air was selected as the baseline for the new instrument based on these simulation results.

Fig. 3. (a) The visibility and sensitivity response to OPD offset. (b) The error of wind retrieval in different OPD offset. Each line is an average of Monte Carlo simulation with 10,000 samples.  $V_I(x)$  is the visibility function as described in Eq. (1).

### 3.3. Field-widening




Fringe contrast in a Michelson interferometer typically decreases at larger field angles, but this issue can be mitigated by applying field-widening technology in an ASHS system. The fringe intensity of on-axis ray is discussed in Eq. (1). When solid angle of the incident illumination is  $\Omega_m$ , the fringe intensity Eq. (1) can be rewritten as

$$I(x) = \Omega_m \int_{-\infty}^{+\infty} B(x) \, d\sigma \left\{ 1 + \operatorname{sinc}\left(\frac{4 \tan \theta \, \sigma x + \sigma \Delta d(\omega)}{2} \Omega_m\right) \right.$$

$$\cos 2\pi \left[ 4 \tan \theta \left(\sigma - \sigma_L - \frac{\Omega_m \sigma}{4\pi}\right) x + \sigma \Delta d(\omega) \left(2 - \frac{\Omega_m}{4\pi}\right) \right] \right\}. \tag{7}$$

When  $\Omega_m = \frac{\pi}{4 \tan \theta \sigma x + \sigma \Delta d(\omega)}$ , the maximum amplitude of I(x) is achieved (Harlander, 1991;

Wei et al., 2020). That means the field of view is limited by the spatial frequency and extended distance  $\Delta d(\omega)$ . The influence of incident light at different angles on spatial frequency can be eliminated by inserting the prisms to enlarge the field of view. This prism can be defined by the minimum deviation angle  $\eta$  and Littrow angle of the gratings  $\theta$ ,

$$2(n^2 - 1)\tan \eta = n^2 \tan \theta,\tag{8}$$

where n is the refractive index of the prism (Harlander, 1991). The apex angle  $\alpha$  of the prisms can be determined by the relationship,

$$n\sin\frac{\alpha}{2} = \sin\eta. \tag{9}$$

It should be noted that the extended distance  $\Delta d(\omega)$  depends on  $\omega$ , the angle between the ray of incident light and the optical axis when considering off-axis light. According to Eq. (7) of Wei et al. (2020), the difference of OPD for rays in different incident angles can be compensated when the higher order term is set to zero, i.e.,

$$\frac{t_1}{n_1} + d_1 - d_2 = 0. (10)$$

The field-widening is achieved when the configuration complies with both Eq. (9) and Eq. (10).

A ray-tracing optimization approach is applied in this study to determine the parameters that

minimize the phase difference between the on-axis and off-axis rays by iterating parameters of prisms, spacers and asymmetric thickness of the beamsplitter.

### 3.4. ASHS interferometer configuration and optimization



The monolithic ASHS interferometer is constructed from different types of glass, each with unique thermal expansion properties, making it sensitive to changes in the ambient environment. The thermal sensitivity of these optical components can lead to phase drift and visibility deterioration in the interferograms. Careful selection and combination of materials minimize this sensitivity to achieve an optimal thermal design, the derivatives of both the phase and OPD with respect to temperature must be minimized:

$$\frac{d\phi(x)}{dT} = 2\pi \left(\frac{df_x}{dT}x + 2\frac{d\Delta d}{dT}\sigma\right) = 0,\tag{11}$$

where T is the temperature,  $f_x = 4 \tan \theta \, (\sigma_0 - \sigma_L)$  is the spatial frequency. This equation holds when the two terms of the derivative,  $\frac{df_x}{dT}$  and  $\frac{d\Delta d}{dT}$ , are equal to zero at the same time (Wei, 2021).

Ray-tracing software is applied to optimize the ASHS interferometer configuration by considering restrictions of the OPD offset, field-widening and thermal compensation, etc. The final optimization parameters for measuring the atomic oxygen green line emission are given in Table 1.

Table 1. Specification of ASHS Interferometer

| Interferometer | OPD offset        | 55 mm                                |
|----------------|-------------------|--------------------------------------|
|                | Spatial frequency | 16.67 cm <sup>-1</sup> (557.7 nm)    |
|                | Gratings          | groove density: 900 mm <sup>-1</sup> |
|                |                   | Material: Fused silica               |

|  |                      | Littrow angle: 14.55°                     |
|--|----------------------|-------------------------------------------|
|  | Beamsplitter         | Material: CDGM H-K9L                      |
|  |                      | Asymmetric Beamsplitter                   |
|  |                      | $t_0 = 50 \text{ mm}$                     |
|  |                      | $t_1 = 23.1 \text{ mm}$                   |
|  | Field-widening prism | Material: Schott N-Lak12                  |
|  |                      | Apex angle: 13.29°                        |
|  |                      | $t_2 = 10.6 \text{ mm}$                   |
|  | Spacers              | JGS1 (Spacer 1),<br>CDGM D-FK61 (Spacer2) |
|  |                      | $d_1 = 8.7 \text{ mm}$                    |
|  |                      | $d_2 = 21 \text{ mm}$                     |

### 4. ASHS instrument configuration




The configuration of the ASHS instrument is depicted in Fig. 4, with component information provided in Table 2. In addition to the ASHS, the complete instrument includes front optics that image the atmospheric scene onto the gratings and camera optics that image the gratings onto the focal plane array, which are designed separately. Within the front optics, a beamsplitter overlays the atmospheric scene with light from a krypton lamp, which is used to monitor instrument drifts. Furthermore, a mirror system is installed between the instrument and the atmosphere, allowing observations in the four cardinal directions and zenith to obtain zero wind measurements.

The M1, M2 mirrors are installed on a two-axis all-sky scanning system to collect airglow radiation in 45° elevation and zenith directions. The telescope L1 to L2 and the aperture stop are designed with a full field of view of 9° to ensure high luminous flux and signal-to-noise ratio of the instrument. The calibration line ( $\lambda = 557.029$  nm) from a krypton lamp is attenuated

by neutral density filter (ND) to an intensity that is comparable to the atomic oxygen green line. The collimated and holographically diffused calibration line is transmitted by a beamsplitter BS1 and enters the relay lens. The airglow and the calibration signal, collimated by relay lenses L4 to L5, both enter the interferometer and produce fringe images. F1 is a narrow band filter with a center wavenumber of 557.7 nm and a band width of 2 nm, so that background radiation and other spectral lines of the krypton lamp are suppressed.

The double telecentric system L6 to L7 with a magnification of 1.0 relays the fringe pattern from the localization plane of the interferometer to the focal plane of the detector. The 2048×2048 pixels weak signal detector with a pixel size of 13.5 µm acquires images in an area of 27.6×27.6 mm. To increase rigidity and reduce system deformation, the interferometer, telecentric system, and detector have been installed on a thick aluminum alloy plate. Additionally, a metal shell has been added to the plate. The components in dashed line in Fig. 4 are integrated into an insulation box made of insulation board filled with aluminum silicate fibers and covered with aluminum film so that the interferometer and optical systems are prevented from being impacted by rapidly changing ambient temperatures.

The interferometer is integrated into an active temperature-controlled housing, which includes an aluminum container, a polystyrene insulating enclosure, a set of heating tapes, a set of temperature sensors, and a PID (proportional-integral-derivative) controller. Enclosure windows W1 and W2 are installed at the entrance and exit positions. Temperature sensing is completed by platinum resistors and thermistors and a target temperature at 30 °C is set and maintained by the PID controller. The whole temperature-control system keeps the temperature inside the aluminum container better than  $\pm$  0.1 °C.

Fig. 4. The configuration of the ASHS instrument. M denotes mirrors, L denotes lenses, F denotes narrow band filters, ND denotes neutral density filters, BS denotes beamsplitters, G denotes gratings, P denotes field widening prisms, S denotes plane parallel spacers, W denotes interferometer enclosure windows.

### **Table 2. Major Parameters of the Instrument Components**

| Instrument components  | Major element                   | Parameter                       |
|------------------------|---------------------------------|---------------------------------|
| Optical system         | Full field of view              | 9°                              |
|                        | Etendue                         | 0.156 cm <sup>2</sup> steradian |
|                        | Relay lens magnification        | 1.0                             |
|                        | Center wavelength of the filter | 557.7 nm                        |
|                        | Band width of the filter        | 2 nm                            |
|                        | Size                            | 2048×2048                       |
| Detector               | Pixel length                    | 13.5 μm                         |
| Calibration illuminant | Illuminant wavelength           | 557.029 nm                      |


Neutral density filter OD 0.8

Fig. 5. The monolithic ASHS interferometer (a) and assembled ASHS instrument (b). All spacers are made into a hollow structure to reduce the thermal expansion.

### 255 **5. Laboratory calibration**


### 5.1. Thermal performance

Although the ASHS was designed to be highly thermally stable, as described in Section 3.4, the as-built configuration may exhibit some unintended thermal sensitivity due to manufacturing tolerances in the components and the assembly process. The use of UV-curing bonding layers, whose thickness is intended to compensate for component tolerances, can affect the thermal performance as well. Therefore, it is indispensable to verify the as-built thermal stability of the interferometer.

Fig. 6. The configuration of thermal performance test.

Fig. 7. The sensitivity of spatial frequency (a) and phase offset (b) when the temperature changes from 25.5  $^{\circ}$ C to 30  $^{\circ}$ C.



Fig. 6 is a setup for thermal performance test based on the instrument configuration in Section 4. The illumination system, temperature-controlled interferometer and imager are encapsulated in an insulation housing, which prevents air exchange with the external environment. The ambient temperature is set at 24 °C through the air conditioning system. The interferometer was thermally regulated using the PID controller within a temperature range of 25.5 °C to 30 °C in 0.5 °C increments. Every four hours, when the temperature was stable,

frames of the interferograms are continuously sampled for half an hour, with each exposure time of 30 seconds. After darkfield, flatfield and phase distortion correction for each interferogram, the phase was extracted by the following equation,

$$\phi(x) = \tan^{-1} \left\{ \frac{\Im[I_D(x)]}{\Re[I_D(x)]} \right\},\tag{12}$$

where  $\Im[I_D(x)]$  and  $\Re[I_D(x)]$  are the imaginary part and real part of the signal-modulated portion, respectively. The phase of the center 1600 pixels was fitted linearly using the least squares method. The slope of the fitted line divided by  $2\pi$  is taken as the spatial frequency and the intercept is taken as the phase offset.

Fig. 7 shows the response of spatial frequency and phase offset to the change of temperature. Each point represents the average of approximately 44 measurements. The solid lines are the result of a linear least squares fitting. The spatial frequency and the phase offset are linearly correlated with temperature, which are  $-0.038 \, \mathrm{cm}^{-1}/^{\circ}\mathrm{C}$  and  $0.624 \, \mathrm{rad}/^{\circ}\mathrm{C}$ , respectively. The thermal sensitivity, in conjunction with a precision temperature control system, restricts thermally induced phase drift to a detectable range. The residual thermal drift can be monitored and compensated through a synchronous calibration line (Harlander et al., 2010).

### 5.2. Doppler velocity calibration





To verify the key as-built instrument parameters and to characterize the instrument prior to field deployment, laboratory calibration is indispensable. Specifically, according to Eq. (5), the OPD may deviate from the design value during manufacturing and UV-curing, which introduces systematic errors in the wind retrieval from phase difference (Zhu et al., 2023).

A Doppler-shift generator is established as shown in Fig. 8. A krypton lamp is used for illumination. The light is focused onto a high-speed rotating disk covered with retroreflective film, which reflects the light back along its original path. This reflected light is then directed by a beamsplitter into the interferometer. A narrow band filter is applied to select krypton emission

line at 557.029 nm entering the interferometer. The disk is driven by an electric motor which is set at 1000-5000 RPM (revolutions per minute) corresponding to a Doppler velocity of 18.6-93.1 m/s. Ten frames of interferograms are recorded for each speed in the experiment. The first ten frames at 1000 RPM serve as a baseline, representing a condition analogous to the zenith overhead wind. This approach ensures that the signal-to-noise ratio (SNR) remains uniform across all frames. After image correction and phase extraction, the phase difference is obtained by subtracting the baseline from an average phase of ten frames of interferogram at different speeds.


Fig. 8. The configuration of laboratory calibration for the ASHS instrument.

Fig. 9(a) Fitting results of the Doppler velocity compared with expectations. Each point is an average of 10 measurements. (b) Retrieval deviation of the retrieved Doppler velocity and the expectations.

Fig. 9(a) shows the wind retrieval using the laboratory measurements mentioned before. The fitting coefficient is 532.2 ms<sup>-1</sup>/rad, which is very close to the designed value 531.9 ms<sup>-1</sup>/rad. The retrieval deviation from generated Doppler speed is shown in Fig. 9(b). In 50 samples, the deviation is distributed from -2 m/s to 1.8 m/s. Due to the lack of calibration line in this experiment, an algorithm is applied to fit the thermal drift estimation curve by continuous sampling and piecewise fitting. This approach is based on the assumption that the trend of thermal drift remains consistent over a short period. The underlying principle involves quantifying the contribution of thermal drift to the phase shift throughout the experiment by fitting the thermal drift curve. Each data group is partitioned into two segments for fitting purposes, as the drift trend varies across segments. The deviation in Fig. 9(b) may be primarily induced by the thermal drift. The standard deviation was 0.95 m/s, and no systematic error was

found.




### 6. First Field observations

Neutral wind measurements were conducted at Mohe Station (53.5°N, 122.3°E) during nighttime on January 14, 2024, when the solar zenith angle exceeds  $100^{\circ}$ . The exposure time for each sample is 5 minutes, and a complete scanning cycle, which includes measurements in the northward, eastward, zenith, southward, and westward directions, takes approximately 26 minutes. The darkfield is taken every night before observation by closing the entrance aperture of the detector. Fig. 10 is an example of a raw image containing information of atomic oxygen green line and calibration line. There are notches marked by laser on the top of one grating, which is used for calibrating the thermal drift outside the interferometer (Marr et al., 2020). After removing the dark field and replacing the hot pixels and spikes, the image is binned  $2 \times 8$ , 2 pixels in x direction and 8 pixels in y direction, resulting in a final resolution of  $1024 \times 256$  elements. In order to avoid large distortions at the interferogram edges, only  $800 \times 150$  elements are used for retrieving wind.

Fig. 10. Raw image, facing westward at a 45° elevation, captured at Mohe Station at 9:12 UT on January 14, 2024.

Fig.11. Neutral wind measured by the ASHS and the co-located LiDAR at Mohe Station during nighttime on January 14, 2024. The black, green and blue lines are obtained by the ASHS detecting in different directions. The red line represents the weighted average result of the LiDAR measurements at an altitude ranging from 90 to 100 km.





Fig. 11 shows the observational results of the geomagnetic quiet night on January 14, 2024 at Mohe station. After removing the outliers, caused by light pollution or detector sampling errors, the average phase in the zenith direction throughout one night is taken as the zero-wind phase. In order to validate the ASHS instrument, the results are compared with measurements from atmosphere wind-temperature-metal-constituents LiDAR at Mohe station. The red line represents a weighted average result of LiDAR measurements. The weighting function is a Gaussian function defined as:

$$y(z) = \frac{1}{\sigma\sqrt{2\pi}} \exp\left(-\frac{(z-z_0)^2}{2\sigma^2}\right),\tag{13}$$

where  $z_0$  is the peak height,  $\sigma$  is the standard deviation, and the Full Width at Half Maximum (FWHM) equals  $2\sqrt{2ln2\sigma}$ . We referred to Liu et al. (2025)'s results, and chose the peak height of 95 km and the FWHM of 12.5 km.

The error bars of the ASHS represent a combination of the uncertainty associated with the

zero wind reference and the retrieved phase. On the one hand, since the vertical wind is used as a zero wind reference, its uncertainty can be represented by the standard deviation of the vertical wind throughout the night. On the other hand, According to Eq. (3), there is a theoretical distribution for the phase of the interferogram, with its value being contingent upon the sampling position x of the detector. By conducting a linear regression analysis on the relationship between the retrieved phase and x, we can derive the fitted phase. The uncertainty of the retrieved phase can be obtained by calculating the root mean square of the fitted phase and the retrieved phase. This combination provides a comprehensive indication of the precision and reliability of the data.






Several primary factors contribute to the occurrence of large error bars in the ASHS instrument observations: Firstly, they manifest at the beginning and end of the observation period, primarily due to the degradation of the SNR of the airglow signal caused by the scattering of sunlight in the atmosphere during sunrise and sunset. Secondly, artificial light pollution in the observational environment or detector sampling errors can also lead to such large error bars. Thirdly, meteorological conditions, such as cloudy or rainy weather, play a role. Additionally, inherent variations in the intensity of the airglow signals themselves are another contributing factor.

During the observation periods, the sky was clear and geomagnetic activities were quiet. After sunset, the meridional wind was directed northward while the zonal wind blew eastward. At about 11:30 UT, the meridional wind abruptly shifted to the southward direction and maintained a velocity of approximately 25 m/s until it steadily reverted to a northward direction from 18:00 UT to 20:00 UT. During the period from 14:00 UT to 16:00 UT, the zonal wind gradually transitioned from an eastward to a westward direction, subsequently reversing back to an eastward flow for the remainder of the observational period.

The meridional and zonal wind measurements from the ASHS and the LiDAR exhibit a

satisfactory consistency in terms of both the temporal synchronization and the magnitude of wind variations. This congruency not only attests to the reliability of the ASHS in capturing wind dynamics but also underscores the comparability of the two distinct measurement systems. Additionally, the consistency of the east-west zonal wind and north-south meridional wind is confirmed by the measurements, as the ASHS instrument's observations in opposite directions corroborate each other. Furthermore, the measured zenith wind remained close to zero throughout the night, indicating strong instrument stability.






To facilitate a comparative analysis of the correlation between data obtained from two distinct systems, it is imperative to standardize the observational outcomes. For this purpose, data from both ASHS and LiDAR systems, collected during overlapping observation periods spanning five days in January 2024. Given the disparate observation frequencies of these systems, piecewise linear interpolation was employed to synchronize their time series. Subsequently, the average horizontal wind component aligned with the north-south axis was designated as the meridional wind, while that aligned with the east-west axis was identified as the zonal wind. The processed time series results, with a five-minute interval, are presented in Fig. 12. The upper and lower panels of the figure illustrate the meridional and zonal wind results, respectively. Within the figure, red dots signify LiDAR observations, whereas blue dots represent observations from the ASHS interferometer. Notably, the observational data from both instruments exhibited good agreement on the nights of the 14th and 15th. However, from the 17th to the 31st, a certain degree of deviation was observed, which may primarily be attributed to inaccuracies in the weighted average calculation of LiDAR detection results across various altitudes. The spatial and temporal variability in the height distribution of oxygen atom OI 557.7 nm airglow necessitates a more precise weighted average computation, based on the specific airglow emissivity height distribution prevalent in the region on the respective day.

Fig.12. The top panel depicts the meridional wind component, while the bottom panel displays the zonal wind component. The blue dots indicate observations recorded by the ASHS, and the red dots represent the weighted average results obtained from LiDAR observations.

Fig.13. Correlation scatter plots of the comparison of neutral wind measured by the ASHS and LiDAR during nighttime in January 2024. The upper left corner indicates the linear fitting equation (y) and the Pearson correlation coefficient (R).

Fig. 13 demonstrates the correlation between the observational outcomes of the ASHS interferometer and LiDAR. The vertical coordinates of the blue dots in the figure represent the observational data from the ASHS interferometer, while the horizontal coordinates correspond to the weighted average of the measurement results obtained from LiDAR. The red line depicted in the figure results from a linear regression fit utilizing the least squares method. 420 The upper left corner of the figure indicates the linear fitting equation (y) with regression slopes ( $k = 0.880 \pm 0.023$  meridional;  $k = 0.812 \pm 0.027$  zonal) and the Pearson correlation coefficient (R = 0.845 meridional; R = 0.780 zonal), both of which highlight a significant positive correlation between the observational results of the two systems. In the linear fitting equation, the slope is observed to be less than 1. As found by Kristoffersen et al. (2024) and 425 Liu et al. (2025) winds measured by different type of optical interferometers are smaller than weighted winds derived from lidars and meteor radars. It may result from that the assumed weighting function has a bias by comparing with the real airglow layer. Noting that in Fig. 13, the slopes of the fitted lines for meridional wind and zonal wind are inconsistent, which may stem from two factors. The field of view and detection scope exhibit differences between 430 ASHS and lidar systems. ASHS detects winds in the four cardinal directions separately and then combines these measurements to derive the meridional and zonal winds. LiDAR measures winds in two orthogonal directions to obtain the meridional and zonal wind components. Under the assumption of a uniformly distributed wind field near the observation region, the slopes of the meridional and zonal winds in Fig. 13 should theoretically be 435 identical. However, during the five-night observation period presented, fluctuations in the wind field distribution near the observation region may have occurred, potentially influenced by small-scale gravity waves, for instance. Another contributing factor could be the assumption of homogeneous airglow volume emissivity distribution in the area during the

weighted - average calculation of LiDAR data. As the actual distribution is asymmetric, it

causes a greater discrepancy between estimated and actual values in one direction.

Consequently, this may result in the zonal fitting slope being lower than the meridional one.

#### 7. Summary





We present a newly developed ground-based ASHS instrument for observing the atomic oxygen green line to derive neutral winds in mesopause region. The temperature of the interference module in the field observatory is kept within ±0.1 °C of the target temperature by utilizing a combination of active and passive temperature control systems. The thermal compensation design results in a spatial frequency thermal sensitivity of 0.038 cm<sup>-1</sup>/°C and a phase offset thermal sensitivity of 0.624 rad/°C. In laboratory measurements, the sensitivity of the DASH instrument on Doppler velocity is calibrated, indicating an uncertainty of less than 2 m/s due to the thermal sensitivity of the instrument. First field measurements taken at Mohe Station were presented, showing a well consistent picture of wind fields with the LiDAR measurements. The regression slopes less than 1 could potentially arise from a systematic underestimation inherent in the ASHS measurement technique or from biases introduced during the process of weighted averaging of wind data across various altitudes. Considering that the present comparative analysis is confined to data collected over a mere five-day period, a comprehensive and in-depth examination of the root causes necessitates the acquisition of extended-duration datasets.

### Data availability

The observational data of DCOI (Dual-Channel Optical Interferometer) and Na lidar in this paper are available at the Chinese Meridian Project (CMP) (<a href="https://www.meridianproject.ac.cn">https://www.meridianproject.ac.cn</a>). The experimental data can be provided by the corresponding authors upon request.

#### **Author contributions**

YZ conceived and led the development; GZ assembled the instrument and wrote the manuscript draft; TW, WL and WY performed the measurements; SL and GY provided LiDAR data; MK and JX reviewed and edited the manuscript.

### 465 Competing interests

The authors declare no conflict of interest.

### Acknowledgments

We thank the Chinese Meridian Project for the support fund. We are also grateful to the anonymous reviewers for their constructive comments and suggestions to improve this manuscript.

### Financial support

This work was supported by the Project of Stable Support for Youth Team in Basic Research Field, CAS (YSBR-018); the National Natural Science Foundation of China (42174212); the Chinese Meridian Project; the Specialized Research Fund for State Key Laboratories.



470

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
