# Peer review of "Development and validation of a ground-based Spatial Heterodyne Spectroscopy Asymmetric (ASHS) system for sounding neutral wind in the mesopause"

_EGUsphere, 2025_

## Author Comment (AC1)

To Reviewer 1

We sincerely appreciate the reviewers' conscientious and meticulous evaluation of our manuscript. Your insightful and crucial suggestions have effectively addressed several oversights in our initial considerations, significantly enhancing the robustness and comprehensiveness of our study. We are truly grateful for your valuable contributions. All the comments from you have been considered in the revised manuscript. I will respond to each of your comments in red font below.

1. How much does the ASHS instrument drift during the Doppler velocity calibration? In the field configuration you have described, a krypton lamp is employed to monitor instrument drift. However, the krypton lamp was used during laboratory calibration to simulate airglow, and no synchronous light source was available to monitor instrument drift. How much does this affect the calibration results?

Reply:

All phase differences and their corresponding Doppler velocities are presented in Fig. 13(a). For the same set of data (at an identical Doppler velocity), the phase shift is predominantly caused by thermal drift, which can lead to substantial retrieval errors. Given the low luminance of the krypton lamp, an exposure time of 30 seconds per frame is necessary, which adequately captures the impact of thermal drift.

Consequently, an algorithm is employed to fit the thermal drift estimation curve through continuous sampling and piecewise fitting. This approach is based on the assumption that the trend of thermal drift remains consistent over a short period. The underlying principle involves quantifying the contribution of thermal drift to the phase shift throughout the experiment by fitting the thermal drift curve. Each data group is partitioned into two segments for fitting purposes, as the drift trend varies across segments. After eliminating the contribution of thermal drift, the results are depicted in Fig. 13(b). It is important to note that the detector operates continuously throughout the experiment, including during the intervals between two groups with different velocities (not illustrated in Fig. 13), as it takes approximately ten seconds for the speed to reach a stable state.

[Figure]

Fig. 13 (a) Continuously sampling and retrieval the phase difference when the Doppler velocity increases from 0 to 74.5 m/s. The solid line is the fitting line of the phase shift contributed to the thermal drift. (b) Phase difference after removing thermal drift contribution. The dash line is the average of the phase difference corresponding to each Doppler velocity.

2. In Fig.11, some of the observed data have very large error bars. This reason must be discussed, as the effective observation sampling rate is also a crucial indicator of an instrument.
Reply:
In the revised version, we have added a discussion on the reason.
Several primary factors contribute to the occurrence of large error bars in the ASHS instrument observations: Firstly, they manifest at the beginning and end of the observation period, primarily due to the degradation of the SNR of the airglow signal caused by the scattering of sunlight in the atmosphere during sunrise and sunset. Secondly, artificial light pollution in the observational environment or detector sampling errors can also lead to such large error bars. Thirdly, meteorological conditions, such as cloudy or rainy weather, play a role. Additionally, inherent variations in the intensity of the airglow signals themselves are another contributing factor.

3. How is the data from LiDAR weighted and averaged?
Reply:

The weighting function is a Gaussian function defined as:

$$y(z) = \frac{1}{\sigma\sqrt{2\pi}} \exp\left(-\frac{(z - z_0)^2}{2\sigma^2}\right)$$

where $z_0$ is the peak height, $\sigma$ is the standard deviation, and the Full Width at Half Maximum (FWHM) equals $2\sqrt{2ln2}\sigma$. We referred to Liu et al. (2025)'s results, and chose the peak height of 95 km and the FWHM of 12.5 km.

We have added the description of the weighting function.

4. The manuscript only compares data from different systems for a few days. Perhaps there should be more comparative data since 2024?

Reply:

Operational constraints limited Mohe station's Na lidar data set to five observational days during its commissioning phase in the early period of 2024. Our objective is to present the development of a ground-based ASHS system, with LiDAR being utilized in this context to validate that the new ASHS instrument is capable of measuring neutral winds in the mesopause. Subsequently, we will continue to conduct wind measurement analysis by integrating more long-term data. For the comparative analysis of data between ASHS and LIDAR as well as meteor radar, please refer to Liu et al. (2025).

---

## Author Comment (AC2)

To Reviewer 2

We sincerely appreciate the reviewers' meticulous review and constructive feedback. Your insightful comments have highlighted key considerations we initially overlooked, significantly enhancing the rigor and clarity of our work. The valuable suggestions have been instrumental in refining our methodology and addressing potential limitations. We are grateful for your time and expertise in elevating the quality of this study. All the comments from you have been considered in the revised manuscript. I will respond to each of your comments in red font below.

Major Comments & Reply

1. There is a lack of sufficient references, notably in sections 2 and 3.2. Please include at least some references in these sections.
Reply:
We have added reference Liu et al. (2018) where the Eq. (1) for interference fringes is presented. We have also included the following references Englert et al. (2023), García-Comas et al. (2008), Hall et al. (2006), Holmen et al. (2016) and Yi et al. (2021) when describing the typical wind and temperature in the mesopause.

2. The intensity is written as a function of the etendue, equation 7, with a max amplitude described for a particular solid angle. Liu et al. (2018) is referenced for this equation, but this equation does not appear in that paper. Please verify and find the correct citation. In addition, it is my understanding that the limitation of large incident angles is that the fringe phase starts to change too rapidly with respect to typical pixel sizes, which causes the reduction in fringe contrast. Field-widening the instrument reduces the phase change, allowing for larger incident angles to be included.
Reply:
Sorry for our mistake, the reference has been replaced (Harlander, 1991; Wei et al., 2020). Light incident at different solid angles of view will form fringes with different phases and spatial frequencies. When these fringes are superimposed, the trend of amplitude follows a sinc function, as shown in Eq. (7). For the specific derivation of Eq. (7), please refer to chapter 5 in Harlander (1991).

3. Line 225: Does having the interferometer attached to a large aluminum plate affect the thermal stability of the system? Although a thermally insulated box would help to reduce temperature changes, the interferometer being attached to the highly thermally conductive aluminum plate would cause the interferometer to be more sensitive to ambient temperature changes.
Reply:
The influence of thermal expansion/shrink of the aluminum plate is unavoidable. To reduce its effect, a double telecentric imaging system was selected, which can keep the interference fringes clear; The interferometer and the metal plate are also soft connected by using four pillars made of polymer materials. The insulation board is filled with aluminum silicate fibers and covered with an aluminum film on its surface. These measures will effectively reduce the impact of thermal expansion of the metal plate. We have adjusted the relevant expressions.

4. Please include uncertainties in Figures 7, 9, 11 (lidar), and 12. When possible, uncertainties should be included when presenting experimental data.

Reply:

Uncertainties have been included in these figures.

5. For the wind wheel experiment, why were thermal drift calibration measurements not included? If it were too difficult to implement a simultaneous calibration line, a thermal drift estimate could be made by alternating 'wind' measurements and thermal drift measurements (e.g. 1000 rpm, 2000 rpm, 1000 , 3000, 1000 etc.). Given the high sensitivity of the interferometer to thermal drift, this must be properly accounted for.

Reply:

It is difficult to purchase a stable frequency light source near 557nm other than a krypton lamp on the current market. We have adopted a method to continuously sample and track the thermal drift of the instrument. The thermal drift of the instrument is reflected by fitting the thermal drift curve (phase change at the same velocity) during the experiment. The details will be presented in the next item.

6. Line 306: It is stated that the thermal drift is estimated using continuous sampling and piece-wise fitting. How do you determine what is attributable to thermal drift and what to Doppler shift? Continuous sampling of what? Temperature? What type of piece-wise fitting? Linear, cubic spline? This process needs to be described in more detail.

Reply:

All phase differences and their corresponding Doppler velocities are presented in Fig. 13(a). For the same set of data (at an identical Doppler velocity), the phase shift is predominantly caused by thermal drift, which can lead to substantial retrieval errors. Given the low luminance of the krypton lamp, an exposure time of 30 seconds per frame is necessary, which adequately captures the impact of thermal drift.

Consequently, an algorithm is employed to fit the thermal drift estimation curve through continuous sampling and piecewise fitting. This approach is based on the assumption that the trend of thermal drift remains consistent over a short period. The underlying principle involves quantifying the contribution of thermal drift to the phase shift throughout the experiment by fitting the thermal drift curve. Each data group is partitioned into two segments for fitting purposes, as the drift trend varies across segments. After eliminating the contribution of thermal drift, the results are depicted in Fig. 13(b). It is important to note that the detector operates continuously throughout the experiment, including during the intervals between two groups with different velocities (not illustrated in Fig. 13), as it takes approximately ten seconds for the speed to reach a stable state.

Since this part of the experimental content has limited relevance to the main theme of the manuscript, we initially did not include it in the main text. In the revised version, based on your feedback, we have supplemented more details of the algorithm.

[Figure]

Fig. 13 (a) Continuously sampling and retrieval the phase difference when the Doppler velocity increases from 0 to 74.5 m/s. The solid line is the fitting line of the phase shift contributed to the thermal drift. (b) Phase difference after removing thermal drift contribution. The dash line is the average of the phase difference corresponding to each Doppler velocity.

7. Line 333: The nightly average provided a zero-wind phase, was the variability of the vertical wind taken into account? Given that the horizontal wind measurements are made at 45°, there would be an equal contribution of the vertical wind. For many of the measurements, the vertical wind would be negligible relative to the horizontal winds, but this is not always the case. Do you expect the zenith variability is associated with vertical wind?

Reply:

During the aurora period, due to strong scattering, the zenith variability may not represent vertical wind. In the case of relatively uniform brightness in the sky, we believe that the zenith variability is associated with vertical wind.

We employed the average phase measured in the zenith direction throughout the entire night as the zero-wind-speed reference. This strategy effectively reduces the influence of vertical wind variations on wind measurements, given that vertical wind fluctuations are typically minimal during periods of geomagnetic quietude. This methodology is also broadly implemented in ground-based passive optical wind measurement techniques. When retrieving the horizontal wind

velocity at a specific instant, we consider both the line-of-sight wind component and the instantaneous vertical wind. According to this approach, significant fluctuations in vertical wind can impact the zero-wind reference, subsequently influencing the accuracy of horizontal wind retrieval.

8. Line 335: What is the weighted average used for the lidar measurements? Are they associated with the lidar uncertainties or nominal airglow layer height? Please provide a complete description of the lidar measurements and weightings.
Reply:
We have added the description of the weighting function.
The weighting function is a Gaussian function defined as:
$$y(z) = \frac{1}{\sigma\sqrt{2\pi}} \exp\left(-\frac{(z-z_0)^2}{2\sigma^2}\right),$$
where $z_0$ is the peak height, $\sigma$ is the standard deviation, and the Full Width at Half Maximum (FWHM) equals $2\sqrt{2ln2}\sigma$. We referred to Liu et al. (2025)'s results, and chose the peak height of 95 km and the FWHM of 12.5 km.

9. Line 337: 'the standard deviation between the fitted phase and the retrieved phase.' This statement is not clear. Please describe the fitted phase and the retrieved phase. Is the fitted phase a thermal drift correction?
Reply:
According to Eq. (3), there is a theoretical distribution for the phase of the interferogram, with its value being contingent upon the sampling position x of the detector. By conducting a linear regression analysis on the relationship between the retrieved phase and x, we can derive the fitted phase. The uncertainty of the retrieved phase can be obtained by calculating the root mean square of the fitted phase and the retrieved phase.

In the revised version, we have added descriptions of the two types of uncertainties represented by error bars.

10. Line 368: It is stated that differences between the lidar and ASHS winds can be attributed to inaccuracies in the weighted average calculation of lidar detection results. What inaccuracies? Do you mean increased uncertainty? This could be verified by including uncertainties in the figures. If you intend to say biases in the lidar, what biases? What are you basing this on? It is a bit unsatisfactory to simply say that when the results differ, the ASHS is correct and the lidar is biased.
Reply:
We do not intend to imply that any instrument's results are inaccurate, nor can such a conclusion be drawn solely from the preliminary comparisons presented in this manuscript. ASHS and LiDAR represent two systems with fundamentally different sensing modalities. LiDAR can measure winds at a single altitude, whereas ASHS captures the integrated wind across the entire airglow emission layer. This discrepancy renders a direct comparison between LiDAR and ASHS measurements unfeasible. Therefore, this manuscript introduces a Gaussian function to perform a weighted averaging of the results of LiDAR. As evident from Liu et al. (2025)'s comparison

process, employing different parameters can lead to significant variations, which are not attributable to measurement uncertainties alone. Our point is that such processing methods may introduce additional errors. Additionally, this manuscript does not aim to discuss how to compare the two instruments more scientifically. Our objective is to present the development of a ground-based ASHS system, with LiDAR being utilized in this context to validate that the new ASHS instrument is capable of measuring neutral winds in mesopause. The current dataset is insufficient for a comprehensive analysis of the systematic biases between the two instruments. For details, please refer to Liu et al., SW, 2025.

11. Figure 12: Please explain the discontinuities in the meridional winds. Are there gaps in the time series? If so, it may be easier to follow if the gaps are left in the figure, as this would not imply that there were such discontinuities. In addition, it would be useful to include the lidar and ASHS uncertainties either as error bars or shaded regions. It is important to include uncertainties in measurements.

Reply:

The x-direction in Figure 12 represents the time series of five independent days. The gaps appear between two different days. We have redrawn this figure to avoid misunderstandings and added error bars.

[Figure]

Fig. 12. The top panel depicts the meridional wind component, while the bottom panel displays the zonal wind component. The blue dots indicate observations recorded by the ASHS, and the red dots represent the weighted average results obtained from LiDAR observations.

12. Line 388: 'In the linear fitting equation for zonal wind, the slope is observed to be less than 1, which may be attributed to potential system biases between the two systems or errors introduced during the weighted averaging of wind measurement results from various altitudes recorded by LiDAR.' I think a full description of the weighted lidar average would be useful here. Please attempt to explain why the lidar winds are, on average, larger than the ASHS. What systematic

biases between the systems? Given that the slope is less than 1, this is not likely a simple bias, but related to a systematic underestimation of the wind by the ASHS. Could this be attributable to a difference in atmospheric scattering of the different wavelengths?

Reply:

We have added a full description of the weighted lidar average in the main text and reply 8. As you may be concerned about, there could be systematic underestimation or overestimation in ASHS measurements. We have added possible reasons. As found by Kristoffersen et al. (2024) and Liu et al. (2025) winds measured by different type of optical interferometers are smaller than weighted winds derived from lidars and meteor radars. It may result from that the assumed weighting function has a bias by comparing with the real airglow layer. It is hard to see that this discrepancy is attributable to a difference in atmospheric scattering of the different wavelengths.

Reference:

[1] Kristoffersen, S. , Ward, W. , & Meek, C. (2024). Wind comparisons between meteor radar and doppler shifts in airglow emissions using field-widened michelson interferometers. Atmospheric Measurement Techniques, 17(13).

[2] Liu, W. J., Zhu, Y. J., Xu, J., et al. (2025). Validation of neutral wind in the mesopause measured by a dual‐channel optical interferometer (DCOI) network of the Chinese meridian project. Space Weather, 23, e2025SW004468.

Minor Comments and Reply

1. Remove 'region' from the title.

Reply:

The title has been changed.

2. Line 30: Change '<' to better than.

Reply:

The symbol has been replaced.

3. Line 45: change 'high etendue device' to 'high etendue'.

Reply:

The words have been changed.

4. Line 60: change 'observation' to 'observations'.

Reply:

The word has been changed.

5. Figure 2: What are the angles of these prisms.

Reply:

The specification of ASHS Interferometer is listed in Table 1. We have added hints in the captions of Figure 2.

6. Please include the proper punctuation after equations. Equations should be treated as part of the paragraph, and punctuated in such a manner.

Reply:

The proper punctuation has been added.

7. Line 146: change 'is to 'are'.

Reply:

The word has been changed.

8. Line 150: 'decreases' should be 'increases'.

Reply:

The word has been changed.

9. Line 170: Please define ω.

Reply:

We have added the definition of ω. ω represents the angle between the ray of incident light and the optical axis when considering off-axis light.

10. Line 179: Is ω the same incident angle as η?

Reply:

We have changed the description of η. η represents the minimum deviation angle. The minimum deviation angle is the smallest possible angle by which a ray of light is deviated after passing through an optical prism. This occurs under a specific geometric condition where the angle of incidence at the first surface of the prism equals the angle of emergence at the second surface, making the light path symmetric inside the prism.

11. Line 184: '... the parameters, which minimizes the phase difference...' should be '... the parameters that minimize the phase difference...'

Reply:

The sentence has been corrected.

12. Line 191: Change 'minimize' to 'minimizes'.

Reply:

The word has been corrected.

13. Line 210: Change '...allowing observations in different directions and enabling zenith direction viewing to obtain zero wind measurements.' to '...allowing observations in the four cardinal directions and zenith to obtain zero wind measurements.'

Reply:

The sentence has been changed.

14. Line 216: place parentheses around ND.

Reply:

The parentheses have been added.

15. Line 217: Change 'on' to 'by'.

Reply:

The word has been corrected.

16. Line 218: change 'lens' to 'lenses'.

Reply:

The word has been corrected.

17. Line 229: change 'impacting' to 'being impacted'.

Reply:

The word has been corrected.

18. Line 230: Change 'environment' to 'temperatures'.

Reply:

The word has been changed.

19. Line 231: Perhaps 'housing' would be better than 'thermostat'

Reply:

The word has been changed.

20. Line 237: remove 'stability'

Reply:

The word has been removed.

21. Line 276: It is stated that 'the temperature sensitivity is sufficiently small that it does not impact the wind data quality', however, this is misleading as a temperature change of 0.1° would correspond to approximately 30 m/s for the green line emission. It is rather that due to this thermal sensitivity, calibration measurements must be synchronously made.

Reply:

When thermal sensitivity exceeds a critical threshold such that thermally induced phase drift span more than one full period, the actual thermal drift becomes indeterminable even with synchronous calibration line.

We have revised the expression to avoid misunderstandings. The thermal sensitivity, in conjunction with a precision temperature control system, restricts thermally induced phase drift to a detectable range. The residual thermal drift can be monitored and compensated through a synchronous calibration line.

22. Line 283: 'system' should be 'systematic'.

Reply:

The word has been corrected.

23. Line 291: It is stated that 'reflection of light significantly decreases when the disk is not rotating.' This statement is unclear. Why would the reflection of light significantly decrease when

the disk is not rotating? The retro-reflective material should still work if the disk is motionless, however, there could be biases introduced if there are imperfections in the retro-reflective material that would be averaged out with a rotating wheel.

Reply:

The concern you raised is something we overlooked. We have conducted some research and may be able to explain this phenomenon as follows:

Retroreflective materials typically enable light to be reflected back at an angle close to the incident direction within a certain range through special structural designs, such as glass microspheres and prisms. Taking glass microsphere-type retroreflective materials used in the manuscript as an example, after light enters the glass microspheres, it undergoes processes like refraction and reflection inside, and finally exits in a direction similar to the incident light. This characteristic allows more light to concentrate in a specific reflection direction when the light is incident at an appropriate angle, thereby enhancing the intensity of the reflected light [1,2].

When the turntable rotates, the angle of the retroreflective material surface relative to the light source and the detector changes continuously. During this process, more light will be incident on the material surface at the optimal angle that satisfies retroreflection. For instance, suppose the light emitted by the light source irradiates the turntable with a certain angular distribution. When the turntable is stationary, only part of the area may have the incident angle of light exactly meeting the optimal conditions for retroreflection, while the light reflection in most areas is relatively scattered, resulting in weak overall reflected light. However, when the turntable rotates, the material surfaces at different positions will sequentially experience various incident angles, and more light can be incident at appropriate angles, making the reflected light concentrate within the receivable range of the detector and thus enhancing the intensity of the reflected light.

To avoid unnecessary misunderstandings, we removed the first half of the sentence in the revised version.

Reference:

[3] Yuan, J., Farnham, C., & Emura, K. (2021). Evaluation of retro-reflective properties and upward to downward reflection ratio of glass bead retro-reflective material using a numerical model. Urban Climate, 36.

[4] Meng, X. (2022). Spectral properties of retro-reflective materials by the experimental measurement. SSRN Electronic Journal.

24. Line 332: How are outliers identified? Is this done using standard deviation?

Reply:

We have added the definition of outliers. The outliers refer to data with an extremely high phase uncertainty. This issue is primarily caused by light pollution or detector sampling errors, which can lead to severe degradation of the fringe patterns.

25. Line 354: The nightly average provided a zero-wind phase, was the variability of the vertical wind taken into account? Given that the horizontal wind measurements are made at 45°, there would be an equal contribution of the vertical wind. For many of the measurements, the vertical

wind would be negligible relative to the horizontal winds, but this is not always the case. Do you expect the zenith variability is associated with vertical wind?

Reply:

During the aurora period, due to strong scattering, the zenith variability may not represent vertical wind. In the case of relatively uniform brightness in the sky, we believe that the zenith variability is associated with vertical wind.

We employ the average phase measured in the zenith direction throughout the entire night as the zero-wind-speed reference. This strategy effectively reduces the influence of vertical wind variations on wind measurements, given that vertical wind fluctuations are typically minimal during periods of geomagnetic quietude. This methodology is also broadly implemented in ground-based passive optical wind measurement techniques. When retrieving the horizontal wind velocity at a specific instant, we consider both the line-of-sight wind component and the instantaneous vertical wind. According to this approach, significant fluctuations in vertical wind can impact the zero-wind reference, subsequently influencing the accuracy of horizontal wind retrieval.

26. Figure11: These wind patterns appear to have a period of approximately 12 hours. Is this attributable to a semi-diurnal tide?

Reply:

Such patterns may be attributed to a semi-diurnal tide. To draw a conclusion, it is necessary to combine it with longer-term observational data.

27. Line 354: It is stated that the zenith wind is close to zero indicating strong instrument stability. Were calibration lines not used for these measurements? Or is this the variability after thermal drift correction? If the latter, this is not really indicative of instrument stability.

Reply:

The wind results depicted in Figure 11 are calibrated. The calibration line is employed to monitor the thermal drift within the interferometer. However, it is incapable of resolving all instrumental stability issues. Unlike the experimental setup illustrated in Figure 8, during field observations, the instrument comprises multiple systems. The scanning system and front optics are required to precisely capture the emission lines from a designated direction and feed them into the interferometer. Similarly, the rear optical system and the detector necessitate a stable temperature environment, as failure to maintain this may result in sub-pixel shifts in the fringe pattern. These factors all exert an influence on wind retrieval, yet they cannot be manifested or accounted for by the calibration line.

28. Line 358: remove 'were utilized'

Reply:

The words have been removed.

29. Line 364: It is stated that black dots in Figure 12 are lidar and red are ASHS, but this is opposite of what is shown in the figure legend. Which is correct?

Reply:

The description in the main text is incorrect. We have corrected the description.

30. Line 398: it is stated that the uncertainty of less than 2 m/s is due to the thermal sensitivity. Although this could be partially attributable to the thermal drift, there would also be contributions related to SNR, line visibility, phase precision.
Reply:
The text described calibration results obtained in the laboratory. By utilizing artificial light sources, a sufficiently high signal intensity can be achieved, and environment noise can be effectively eliminated within the laboratory environment. Under such conditions, the impacts of SNR, visibility and phase precision on the results are significantly lower than the order of magnitude of 1 m/s.

31. Line 400: For the lidar comparisons, please be more specific and quantitative.
Reply:
We have added a more specific description in the conclusion. A regression slopes less than 1 could potentially arise from a systematic underestimation inherent in the ASHS measurement technique or from biases introduced during the process of weighted averaging of wind data across various altitudes. Considering that the present comparative analysis is confined to data collected over a mere five-day period, a comprehensive and in-depth examination of the root causes necessitates the acquisition of extended-duration datasets.

---

## Author Response (AR2)

**#1 Reviewer**

We sincerely thank the reviewers for taking the time to give our manuscript such a careful and detailed evaluation. Your insightful and really important suggestions have helped us fix several mistakes we didn't notice at first. This has made our study much stronger and more complete. We're so grateful for all the great help you've given us!

**Comments & Reply**

1. Fig 4 and Fig 6 show the parallel incident illumination, but the beams are not parallel anymore after the telecentric systems and in front of the detector. To my knowledge, it should be parallel if the telecentirc systems relay the fringe localization plane onto the detector. Although there is a diffuser in the setup, it only shows a parallel view after the diffuser in the figures.

**Reply**

Your concern is indeed valid. We have made the necessary modifications in Figures 4, 6, and 8. Once the parallel light traverses the diffuser, it will exhibit a certain degree of divergence. The influence of the diffuser is not depicted in the provided figure. Within this configuration schematic diagram, the representation lacks the stringent precision typically found in a light path diagram.

2. Figure 4 contains numerous abbreviations, such as ND, G, P, S, etc. To facilitate understanding, could you provide the explanations of those abbreviations directly after the figure caption?

We have supplemented the explanations after the figure caption.

**#2 Reviewer**

We would like to express our heartfelt gratitude to the reviewers for their diligent and thorough assessment of our manuscript. Your perceptive and pivotal recommendations have successfully rectified a number of oversights in our initial conceptualization, thereby substantially elevating the rigor and inclusiveness of our research. We are immensely thankful for your invaluable input.

**Comments & Reply**

1. In Figure 2, it is stated that 'The specification of the ASHS Interferometer is listed in Table 1.' First, I think this should be 'The specifications of the ASHS interferometer are listed in Table 1.' Second, I do not see these specifications, e.g., lengths of components (t and d) in Table 1. Please include these values.

**Reply:**

We have corrected the word and supplemented the specifications in Table 1.

2. Figure 12: the x-axis label is clipped and not fully visible.

**Reply:**

The figure has been corrected.

3. Figure 13: the meridional and zonal winds have different slopes (when accounting for the uncertainties), do you have a physical explanation for this behaviour. Please comment on this. Reply:

We have supplemented the comments after Figure 13: Noting that in Fig. 13, the slopes of the fitted lines for meridional wind and zonal wind are inconsistent, which may stem from two factors. The field of view and detection scope exhibit differences between ASHS and lidar systems. ASHS detects winds in the four cardinal directions separately and then combines these measurements to derive the meridional and zonal winds. LiDAR measures winds in two orthogonal directions to obtain the meridional and zonal wind components. Under the assumption of a uniformly distributed wind field near the observation region, the slopes of the meridional and zonal winds in Fig. 13 should theoretically be identical. However, during the five-night observation period presented, fluctuations in the wind field distribution near the observation region may have occurred, potentially influenced by small-scale gravity waves, for instance. Another contributing factor could be the assumption of homogeneous airglow volume emissivity distribution in the area during the weighted - average calculation of LiDAR data. As the actual distribution is asymmetric, it causes a greater discrepancy between estimated and actual values in one direction. Consequently, this may result in the zonal fitting slope being lower than the meridional one.

4. Finally, just a comment, though not something that I think necessarily needs to be addressed in this manuscript, but did you consider the thermal drift when decreasing the temperature. Essentially, do you expect/observe any hysteresis in the thermal drift?

**Reply:**

If you are referring to the experiment described in Section 5.1, we conducted it by gradually increasing the temperature in a controlled manner, as this approach offers better controllability

compared to reducing the temperature. Once the sensor readings reached the target temperature, the instrument was allowed to stabilize for nearly 4 hours. Subsequently, the instrument's parameters were measured. Therefore, no "hysteresis" was observed in our experiment. The phase offset and spatial frequency characterized in this experiment precisely represent the primary effects of thermal drift. Through this methodology, we evaluated the extent to which thermal drift influences wind measurements.